# Short- and Long-Term Outcomes of Preeclampsia in Offspring: Review of the Literature

**DOI:** 10.3390/children10050826

**Published:** 2023-05-01

**Authors:** Sevasti Koulouraki, Vasileios Paschos, Panagiota Pervanidou, Panagiotis Christopoulos, Angeliki Gerede, Makarios Eleftheriades

**Affiliations:** 1Second Department of Obstetrics and Gynaecology, Aretaieio Hospital, National and Kapodistrian University of Athens, 115 28 Athens, Greece; 2Unit of Developmental and Behavioral Pediatrics, First Department of Pediatrics, School of Medicine, National and Kapodistrian University of Athens, 115 27 Athens, Greece; 3Department of Obstetrics and Gynecology, Democritus University of Thrace, 691 00 Campus, Greece

**Keywords:** preeclampsia, offspring, neurodevelopment, fetal cardiac remodeling, obesity, high blood pressure, neonatal sepsis, neonatal RDS, necrotizing enterocolitis, endocrine morbidity

## Abstract

Preeclampsia is a multisystemic clinical syndrome characterized by the appearance of new-onset hypertension and proteinuria or hypertension and end organ dysfunction even without proteinuria after 20 weeks of pregnancy or postpartum. Residing at the severe end of the spectrum of the hypertensive disorders of pregnancy, preeclampsia occurs in 3 to 8% of pregnancies worldwide and is a major cause of maternal and perinatal morbidity and mortality, accounting for 8–10% of all preterm births. The mechanism whereby preeclampsia increases the risk of the neurodevelopmental, cardiovascular, and metabolic morbidity of the mother’s offspring is not well known, but it is possible that the preeclamptic environment induces epigenetic changes that adversely affect developmental plasticity. These developmental changes are crucial for optimal fetal growth and survival but may lead to an increased risk of chronic morbidity in childhood and even later in life. The aim of this review is to summarize both the short- and long-term effects of preeclampsia on offspring based on the current literature.

## 1. Introduction

According to the World Health Organization (WHO), gestational hypertensive disorders complicate up to 10% of pregnancies globally. Preeclampsia, the more severe clinical manifestation of these disorders, affects 3 to 8% of pregnancies worldwide and is a major cause of maternal and perinatal morbidity and mortality, accounting for 8–10% of all preterm births [1]. Preeclampsia is a multisystemic progressive disorder typically presenting after 20 weeks of gestation or postpartum. It is defined as the occurrence of new-onset hypertension along with evidence of maternal organ failure, which may include new-onset proteinuria at ≥300 mg/24 h, hematological complications, deranged biochemical markers of coagulation and hepatic function, or neurological complications and/or evidence of uteroplacental dysfunction such as fetal growth restriction [2] (Table 1).

Based on gestational age, preeclampsia is categorized as early- and late-onset. Early-onset preeclampsia, which develops before 34 weeks of gestation, is less common (10% of the cases) and usually associated with an unfavorable outcome. It mainly affects primigravidae with a normal body mass index (BMI). This disease is usually associated with a severe clinical pattern with a less favorable maternal outcome and concomitant fetal complications such as fetal growth restriction (FGR), demise due to placental abruption, and iatrogenic prematurity [6]. Late-onset preeclampsia, which develops after 34 weeks of gestation, accounts for 90% of the preeclamptic pregnancies and is more often observed in pregnant individuals with risk factors such as chronic hypertension, thrombophilia, an increased BMI, and preexisting diabetes mellitus and autoimmune diseases. Maternal symptoms usually follow a moderate clinical course, with milder complications for the fetus. However, even late-onset preeclampsia can be associated with a severe perinatal outcome [6]. The severe end of the preeclampsia spectrum has been characterized as “preeclampsia with severe features” and entails increased blood pressure ≥ 160/110 mmHg, thrombocytopenia <100.000 platelets/μL, a level of liver transaminases that is at least twice the upper limit of normal concentrations, severe persistent right upper quadrant pain, visual disturbances (photopsia, scotomata, cortical blindness, and retinal vasospasm), persistent severe headaches, and/or acute pulmonary edema [7].

### Pathophysiology of Preeclampsia

Although there are many theories regarding the pathogenesis of preeclampsia, its exact etiology remains unknown and seems to be a multifactorial condition involving fetal/placental and maternal factors [8]. The most prominently accepted theory concerns the abnormal development of the placental vasculature in early pregnancy due to defective trophoblast differentiation and invasion [8], causing abnormal remodeling of the spiral arteries, which leads to the perpetuation of a high-resistance and low-flow uteroplacental unit [9]. In turn, this results in diminished placental perfusion, hypoxia, and ischemia, which lead to the release of antiangiogenic factors into the maternal circulation, an imbalance between vasodilating and vasoconstrictive factors, increased vascular reactivity, and excessive inflammation [10]. The immunologic theory supports the hypothesis that immunologic intolerance between the mother and the paternal/fetal antigens may contribute to the pathogenesis of preeclampsia. This immunologic imbalance is believed to increase natural killer (NK) cell activity and decrease levels of regulatory T cells and other mediators of the immune response, thus inducing abnormal placental implantation [10]. Finally, numerous other theories cast the following features as etiologic or triggering factors: genetic factors, environmental and maternal susceptibility factors (such as IVF, high maternal BMI), increased sensitivity to angiotensin II, complement activation disorders, and endothelial cell dysfunction [8].

Preeclampsia is a multisystemic disorder that is associated with increased short-term maternal and fetal morbidity and mortality. The mother is at increased risk of multisystemic dysfunction and obstetrical complications, while the fetal complications include growth restriction, prematurity, and stillbirth [11]. Currently, the most effective management method for preeclampsia is the delivery of the placenta and fetus [11]. It is now well-established that preeclampsia is also associated with long-term maternal sequelae. Women with a history of a preeclamptic pregnancy [12] have an approximately two-fold increased risk of suffering from a composite adverse cardiovascular outcome and cardiovascular death, a two-fold increased risk of cerebrovascular disease, an up to four-fold increased risk of hypertension and metabolic syndrome, and a more than two-fold increased risk of developing type 2 diabetes and dyslipidemia during the decades following pregnancy [12]. In terms of maternal cardiovascular disease, it remains unclear whether this ailment is directly accelerated by a preeclamptic pregnancy or if common risk factors contribute to both preeclampsia and cardiovascular disease development. An increasing body of evidence suggests that preeclampsia may also affect the offspring’s health independently of prematurity. The mechanism through which preeclampsia increases the risk of the cardiovascular, metabolic, and neurodevelopmental morbidity of the offspring is not well known [13]. It has been suggested that an unfavorable intrauterine environment such as that in preeclampsia induces epigenetic changes that adversely affect developmental plasticity [14]. These developmental changes are crucial for optimal fetal growth and survival but may lead to an increased risk of chronic morbidity in childhood and even later in adulthood [15].

The aim of this review is to study both the short- and long-term effects of preeclampsia in offspring (Table 2, Figure 1).

## 2. Outcome in Offspring

### 2.1. Neurodevelopment

There is evidence that Hypertensive Disorders of Pregnancy (HDP) and Preeclampsia (PE) have been associated with adverse outcomes and an increased risk of neurodevelopmental disorders in the mother’s offspring. Neurodevelopmental disorders (NDs) are a group of conditions with an onset in a child’s developmental period and comprise Autism Spectrum Disorder (ASD); intellectual disability (ID); Communication Disorders; Attention-Deficit/Hyperactivity Disorder (ADHD); Neurodevelopmental Motor Disorders, including Tic Disorders; and Specific Learning Disorders. All NDs have a strong genetic background, as they are associated with environmental parameters that affect early brain development. A broad range of environmental perinatal parameters may affect neurodevelopment, including prematurity, low birthweight, and maternal exposure to certain environmental contaminants and/or drugs, alcohol, or tobacco [16,17]. Hypertensive disorders of pregnancy and preeclampsia are leading causes of several obstetric complications related to adverse neurodevelopmental outcomes. These complications mainly include preterm birth and fetal growth restriction, both of which are considered well-recognized perinatal risk factors for neurodevelopmental disorders [18,19].

However, hypertensive disorders of pregnancy and preeclampsia have been relatively recently added to these perinatal risk factors (independently of their effects on gestational age and birth weight) [20].

Indeed, the “preeclamptic” environment may alter the structure and function of the central nervous system and affect fetal brain development [21]. Although the exact mechanisms associating PE and neurodevelopmental outcomes are still under investigation [22,23], it has been postulated that both inflammation and oxidative stress, the two main pathophysiological mechanisms of PE, may contribute considerably to this association. Potentiated inflammatory processes and oxidative stress affect maternal, placental, and fetal circulation in PE and expose the fetus to both maternal immune activation and increased concentrations of pro-inflammatory cytokines. The fetal brain is directly exposed to deleterious factors that adversely affect neuronal development.

The final impacts of these changes are the interaction of hypoxia and placental ischemia, oxidative stress, angiogenic and growth-factor-related changes, and inflammation. Changes of the neuroanatomy and the cerebrovasculature [9,24,25] may be associated with a higher risk of developing all NDs [26].

#### 2.1.1. Autism Spectrum Disorder (ASD)

According to DSM-5, Autism Spectrum Disorder (ASD) refers to a group of neurodevelopmental disorders characterized by deficits in social communication and limited interest or repetitive behaviors [27]. ASD affects approximately 1–1.5% of children worldwide [28,29]. ASD has a strong genetic background, but environmental parameters have also been identified as risk factors. These environmental risk factors are mostly related to the perinatal period of life since this period is particularly sensitive with respect to brain development. In addition to other perinatal risk factors, such as prematurity and low birthweight, PE has been recognized as an independent risk factor for ASD in several cohort studies [30,31,32], although the evidence from case-control studies is less conclusive [23,33,34]. In the last decade, four meta-analyses concluded that PE is related to a significantly increased relative risk or odds ratio for ASD, ranging from 1.32 to 1.50.

Interestingly, a recent population-based retrospective study suggested an intergenerational association between PE exposure and ASD and ADHD, as it was found that the offspring of preeclamptic mothers who had also been birthed through preeclamptic pregnancies are more likely to be diagnosed with ASD or ADHD than those birthed by preeclamptic mothers without a similar family history [35].

The mechanisms whereby PE increases the risk of ASD are not fully understood. PE induces hypoxia, reducing both global placental histone acetylation and acetyl-CoA. Excessively high levels of placental genome domains increase the levels of histone modifiers [33]. Moreover, the severity of the symptoms may be affected if factors such as birthweight, gestational age, and maternal/gestational health are accounted for [35].

#### 2.1.2. Attention-Deficit/Hyperactivity Disorder (ADHD)

Attention-Deficit/Hyperactivity Disorder (ADHD) is the most prevalent neurodevelopmental disorder, affecting approximately 5–8% of children [36]. Recent prospective, population-based evidence showed that there is a substantial association between maternal preeclampsia and ADHD [37]. A meta-analysis of nine studies (some of them including controls for birthweight and gestational age) reported an odds ratio of 1.31 for childhood ADHD in children exposed in utero to preeclampsia [20]. Females commonly present lower rates of ADHD; however, they may be at an increased risk when presenting a background of prenatal preeclampsia exposure [38]. Further analysis of the risks to males and females is needed to better understand these differential impacts [37].

#### 2.1.3. Intellectual Disability (ID) and Cognitive Function

Cognitive function is an individual’s capacity to adequately think, learn, and remember and is typically measured via standardized psychometric tests. Various studies have depicted a decline in cognitive function among children of preeclamptic mothers compared to controls [39]. In infancy, the mental developmental index (MDI), which is measured via Bayley Scales, is a common tool for the evaluation of an infant’s current level of cognitive development. Three studies have presented poorer MDI scores among exposed offspring in addition to a lower IQ among PE children from 3 to 18 years old [40,41,42].

The association between PE and Intellectual Disability (ID) is still unclear. It is association with intellectual disability and cognitive impairment may be the result of confounding due to shared familial characteristics [26]. The mechanisms that are responsible for these outcomes have not been fully elucidated. Inflammation and oxidative stress are features that are present in a preeclamptic environment and most likely affect neurodevelopmental programing [23].

#### 2.1.4. Cerebral Palsy (CP)

CP describes a range of motor disabilities with a cerebral origin [43]. Although it is directly related to perinatal life, it is not included in the group of NDs. One large retrospective, population-based cohort study revealed the gestational-age-dependent impact of PE on the risk of developing CP. More specifically, the association between PE and CP risk was found to be negative for infants born at 23–31 weeks of pregnancy, nonexistent at 32–36 weeks, and positive for infants born after 37 weeks [44]. A recent meta-analysis and systematic review showed that PE is not associated with CP (independently of gestational age) [45].

#### 2.1.5. Psychiatric Disruptions

Several psychiatric disorders have been associated with PE. A difficult infant temperament (odds ratio: 2.17), which may represent a preceding developmental state of anxiety and mood disorders, is more common among the offspring of mothers with PE [46]. The reason for this outcome is not fully understood, but the preeclamptic environment changes the placental production and metabolism of serotonin, thus jeopardizing placental vascular health and fetal growth [47,48]. Moreover, these children are at a higher risk of depression, independently of age, birthweight, gestational age, and other familial factors [39]. Finally, there is a risk for developing schizophrenia among adults born through preeclamptic pregnancies (with an adjusted odds ratio of 1.3, which increases to 2.0 among preterm births) [49]. Tumor necrosis factor (TNF)-α, which is found at increased levels in the maternal circulation during PE, is known to affect the neuron–microglia crosstalk and glial regulation of synaptic processes relevant to schizophrenia pathophysiology [50,51]. Overall, the data support the notion that there may be an association between PE in pregnancy and increased risk for schizophrenia, although this is still uncertain.

### 2.2. Eye Disorders

Preeclampsia is associated with an increased risk of ophthalmic morbidity. A population-based cohort analysis compared the risk of long-term ophthalmic morbidity among children who had been born via a preeclamptic pregnancy and those who had not. The results showed a significant association between severe preeclampsia or eclampsia and the risk of long-term, vascularly associated ophthalmic morbidity in offspring (no preeclampsia 0.3%, mild preeclampsia 0.2%, and severe preeclampsia or eclampsia 0.5%, *p* = 0.008). However, there is insufficient evidence regarding the association between eye disease and mild preeclampsia [52]. The underlying mechanisms by which PE affects the eyes of newborns are not well understood. It is possible that abnormal placentation results in placental hypoperfusion and hypoxia, which, in turn, intensifies the expression of hypoxia-inducible factor-1 (HIF-1), a transcription factor that is causally associated with systemic endothelial dysfunction in the fetus and the mother, which affects the visual system [53,54,55]. Intrauterine stress related to PE is an additional contributing factor that triggers fetal adaptive epigenetic reprogramming, thereby increasing the susceptibility of a child to vascular diseases later in life, through a permanent alteration of gene expression [56,57,58,59]. Furthermore, preeclampsia-associated iatrogenic preterm birth and may lead to short-term visual morbidity, such as retinopathy of prematurity (ROP) [60,61].

### 2.3. Immune System and Susceptibility to Infections

It has been suggested that abnormal immune function may contribute to the pathophysiology and clinical presentation of PE. Furthermore, it has been proposed that PE reflects an exaggerated maternal inflammatory response to pregnancy and is associated with immune processes similar to organ rejection after allograft transplantation and in graft-versus-host diseases (GVHD) [62].

The impact of preeclampsia on offspring immune system is still being researched. The preeclamptic environment and the way that Trained Immunity (TI), which includes natural immune memory, has been linked with PE are still under investigation. Two major mechanisms have been proposed to associate the deviation from the normal immunological signature of pregnancy with future disease: the epigenetic reprogramming of the gametes following the exposure of the parent to inflammation and the uteroplacental unit through which TI is transmitted from mother to offspring [63]. Since TI and the effects of a complicated prenatal environment persist across generations, it has been suggested that TI could be causally linked to the increased risk of offspring disease following their exposure to excessive in utero inflammation associated with PE.

It has been shown that PE increases the risk of both allergic and severe atopic sensitization and an increased incidence of asthma [9,64,65]. Furthermore, it has been suggested that there is an increased prevalence of neonatal sepsis in both term and preterm infants born to preeclamptic pregnancies based on inflammation and immune dysfunction associated with PE. Neonatal sepsis is the fifth leading cause of neonatal death. Shane et al. [66] define neonatal sepsis as “a systemic condition of bacterial, viral, or fungal (yeast) origin that is associated with hemodynamic changes and other clinical manifestations and results in substantial morbidity and mortality.” Various known risk factors for the development of neonatal sepsis have been described, such as a high maternal BMI, preexisting diabetes, smoking, numerous digital exams, GBS positivity, chorioamnionitis, the use of antibiotics, administration of steroids for fetal lung maturity, and cesarean delivery. A population-based cohort analysis showed that PE is an independent risk factor for the diagnosis of neonatal sepsis until the gestational age of 37 weeks [67]. This association did not persist when the analysis involved only the subgroup of individuals that delivered at term [68].

The way that preeclampsia influences the long-term susceptibility to infections of offspring is still being researched. Two main independent risk factors, which may affect susceptibility to infections, have also been proposed: gestational age at birth and cesarean section as a choice of the delivery [65,69,70]. Preeclampsia influences the development of the offspring’s immune system, either independently or due to complications that arise from the preeclamptic environment, such as iatrogenic prematurity and increased cesarean section rates [9,67]. Further understanding of the mechanisms by which PE affects the offspring’s immune system and predisposes a child to infections is essential to facilitate clinical interventions to reduce morbidity linked with preeclampsia.

### 2.4. Gastrointestinal Diseases

#### 2.4.1. Neonatal Age

Necrotizing enterocolitis (NEC) is a pediatric gastrointestinal disease that is primarily associated with prematurity and low birth weight. The pathophysiology of NEC is multifactorial, for which prematurity, intestinal immaturity, hypoxia, formula feeding, and colonization with pathogenic bacteria are the main risk factors [71,72]. A prospective study showed that NEC incidence among premature infants was significantly higher in those born through preeclamptic pregnancies compared to normotensive ones. Additionally, NEC occurred significantly earlier [28] and lasted considerably longer in premature infants born to preeclamptic mothers compared to those born to normotensive mothers [53,73].

#### 2.4.2. Childhood

Pediatric gastrointestinal diseases in offspring requiring hospitalization are associated with severe preeclampsia or eclampsia as independent risk factors. These diseases include esophageal morbidity, hernias, and functional colonic morbidity. It is hypothesized that within the context of an unfavorable in utero environment, fetal intestinal perfusion might be slightly reduced to preserve blood flow to other vital organs, leading to consequent ischemic bowel injury that may only be notable later in life [74]. However, this result was derived from a large population-based study, and further research based on community databases is warranted, which should focus on genetic backgrounds and environmental exposure during childhood.

### 2.5. Cardiovascular System

In 1993, the Barker hypothesis was developed, which proposed that maternal hypertension or placental ischemia increased the risk of hypertension, cardiovascular disease (CVD), and stroke in the affected mother’s offspring [75]. This theory is now supported both experimentally and epidemiologically [25]. A multifactorial interaction of different mechanisms including genetic background and environmental parameters [76] may explain this association.

#### 2.5.1. Neonatal Age– Early Childhood

The maturation of fetal myocardial cells is influenced by hormone-mediated regulation and the hemodynamic load in utero [77]. Preeclampsia is associated with increased fetal cardiac afterload due to an increase in placental vascular resistance, which promotes early asymptomatic changes to the fetal heart, through the abnormal accelerated maturation of cardiomyocytes [78,79]. This leads to abnormal hypertrophy and altered anatomy in the cardiac components. In preeclamptic pregnancies, the fetal heart has been reported to present an increased size (a median of 0.27 in uncomplicated pregnancies, which can be compared to 0.31 in cases of fetal growth restriction (FGR), 0.31 in cases of preeclampsia with a normally grown fetus, and 0.28 in cases of preeclampsia with FGR; *p* < 0.001), ventricular hypertrophy (measured wall thickness of 0.55 in uncomplicated pregnancies, which can be compared to 0.67 in cases of FGR, 0.68 in cases of preeclampsia with a normally grown fetus, and 0.66 in cases of preeclampsia with FGR; *p* < 0.001), and an increased myocardial performance index. This degree of fetal cardiac remodeling seems to be similar between preeclampsia and fetal growth restriction [78,80]. Still, gestational hypertension is independently correlated with an increase in right ventricular mass during the first three postnatal months [1], while reduced left ventricular longitudinal peak systolic strain has been observed in preterm individuals birthed by preeclamptic mothers when compared to preterm, normotensive ones. The adverse changes in left and right ventricular structure and function seem to be independent of preterm birth. In addition, it has been reported that neonates of preeclamptic mothers present coronary dilatation [81] at birth as well as higher cord blood levels of blood N-terminal pro-B-type natriuretic peptide (NT-proBNP), troponin I, homocysteine, and endothelial vascular cell adhesion molecule-1 expression, presenting early endothelial inflammation and cardiac cell damage. Muñoz-Hernandez et al. [82] have demonstrated that the levels of endothelial colony-forming cells (ECFCs) from cord blood were lower in preeclamptic pregnancies both in preterm and term groups than the normotensive controls. Distinctive postnatal microvascular remodeling was further identified by Yu [83], who observed a loss in total dermal microvascular density over the first three months of life in the offspring of hypertensive pregnancies [81]. Additionally, early-onset preeclampsia has been shown to affect osteoprotegerin concentrations at birth, thus altering the osteoprotegerin–RANKL axis involved in fetal cardiovascular “programming” [84]. These observed cytologic, anatomical, and hemodynamic changes are expressed clinically even in the first month of life. Another study on the offspring of early-onset preeclamptic pregnancies during their first postnatal month showed significantly higher systolic (SBP), diastolic (DBP), and mean blood pressure (MBP) levels among these infants from the second day up to four weeks of life (*p* < 0.001–0.033), thereby characterizing preeclampsia as a determining factor of alterations in SBP, DBP, and MBP during the first postnatal month (F = 16.2, *p* < 0.001; F = 16.4, *p* < 0.001; and F = 17.7, *p* < 0.001, respectively) [85]. Moreover, the use of echocardiography on the offspring of hypertensive pregnancies at the age of 3 months showed a significantly greater left and right ventricular mass index in addition to a smaller right ventricular ejection fraction [80]. Zhou et al. showed a diastolic impairment of the left ventricle in fetuses of preeclamptic pregnancies with or without FGR, especially when complicated with preterm delivery before 34 weeks [86]. Left ventricular diastolic dysfunction was also observed during the first postpartum week in premature infants born to preeclamptic mothers [87]. An echocardiographic study on children between 5–8 years old also reported elevated heart rates and late diastolic velocity at the mitral valve attachments [88]. Furthermore, a common underlying pathophysiological mechanism has been hypothesized between maternal preterm preeclampsia and congenital heart defects (CHD) in the offspring [89,90,91]. Possible candidate mechanisms with a predominantly maternal origin include endothelial impairment, angiogenic imbalance, poor angiogenesis, smooth muscle abnormalities, and subclinical metabolic disorders. Infants exposed to early-onset preeclampsia had greater reported prevalence than the late-onset group, including tetralogy of Fallot, atrioventricular septal defects, valvar dysfunction, and patent ductus arteriosus. Moreover, increased intima-media thickness of the abdominal aorta in neonates born to preeclamptic pregnancies has been reported [92]. Finally, a recent systematic review and meta-analysis by Hoodbhoy et al. on the impact of maternal preeclampsia and hyperglycemia on the cardiovascular system of the offspring [93] reported lower birth weight (MD: −0.41 kg) but increased systolic (MD: 2.2 mmHg) and diastolic blood pressure (MD: 1.41 mmHg) in children under 10 years old born to preeclamptic pregnancies compared to controls.

#### 2.5.2. Adolescence–Early Adulthood

In preeclampsia, vasculotoxic factors that enter the placenta provoke excessive hypoxic pulmonary hypertension and may lead to a premature cardiovascular deficiency through the permanent impairment of the systemic and pulmonary circulation [94]. In a metanalysis of eighteen studies by Davis et al. [95] examining traditional cardiovascular risk factors in children and adolescents exposed to preeclampsia, in utero exposure to preeclampsia was associated with a 2.39 mmHg higher systolic and a 1.35 mmHg higher diastolic blood pressure and an increased Body Mass Index (BMI) by 0.62 kg/m^2^ during childhood and early adulthood. The findings were independent of gender and birth weight. The documented elevation in systolic blood pressure was related to an increase of 8% in mortality via ischemic heart disease and 12% from stroke [25]. Additionally, a 2.5-fold higher risk of scores above the 75th centile of global lifetime risk (QRISK) was observed in young adults exposed in utero to preeclampsia [96]. A causative link between in utero exposure to preeclampsia and metabolic adverse effects has also been studied. According to a recent population-based study, the incidence of obesity was higher in the preeclampsia-affected offspring (odds ratio = 1.34) during early childhood even though the incidence of low birth weight (LBW) was higher [97]. A recent meta-analysis studying offspring BMI during peripubertal life reported a higher risk of obesity (odds ratio 2.12 [1.70, 2.66]; *p* < 0.00001) and increased waist circumference (MD 1.37 cm [0.67, 2.06]; *p* = 0.0001) in the preeclamptic compared to non-preeclamptic group. However, offspring BMI was inversely associated with maternal age in both groups [23]. Studies have also described increased adiposity and expression of obesity-related genes as well as increased placental leptin synthesis and leptin concentration in the cord-blood of offspring prematurely born to preeclamptic mothers [94,98,99]. Moreover, the aforementioned concentric heart remodeling with the hypertrophic ventricles and the reduced left ventricular end-diastolic volume has been observed in adolescents born preterm to preeclamptic mothers [100]. Although prematurity is linked to higher blood pressure in childhood and young adulthood, the endothelial dysfunction observed in the offspring of preeclamptic mothers is unique. Furthermore, siblings of these individuals, born at term from uncomplicated pregnancies, have not presented any of the aforementioned cardiovascular defects, suggesting an effect related to preeclampsia rather than a shared genetic background [101]. Lazdam et al. [102], who compared children exposed in utero to early and late-onset preeclampsia between 6 and 13 years of age, described higher systolic blood pressure and other specific adverse blood pressure characteristics in the first group, which were not seen in late-onset and in normotensive pregnancies.

Furthermore, a population-based study comparing the long-term cardiovascular morbidity in offspring born preterm and at term showed that in the preterm group, both severe and mild preeclampsia had no association with cardiovascular morbidity [57]. On the contrary, preeclampsia was characterized as an independent risk factor for cardiovascular disease in offspring born at term [97]^.^ Another large, population-based cohort study on the effect of maternal hypertension on cardiovascular disease during childhood, adolescence, and young adulthood [103] reported a positive correlation between the timing of the onset and the severity of preeclampsia, with a hazard ratio of 1.48 (CI, 1.30 to 1.67; *p* < 0.001) for early-onset and severe preeclampsia.

A recent meta-analysis by Andraweera [104] et al. of thirty-six studies, evaluating evidence for increased cardiovascular disease risk factors in children exposed prenatally to preeclampsia, demonstrated that in utero exposure was associated with 5.17 mmHg higher mean systolic pressure, 4.06 mmHg higher mean diastolic blood pressure, and 0.36 kg/m^2^ higher mean BMI during childhood and early adulthood. No significant association between exposure to preeclampsia in utero and other CVD risk factors was observed. Davidesko et al. [98] demonstrated that there is a significant linear relationship between the severity of preeclampsia and the development of hypertension in the offspring. Although the measured LDL, HDL, cholesterol, and triglyceride levels were higher in the cord blood in the offspring of preeclamptic mothers, there was no difference found in lipid levels in later years, suggesting a possible temporary impact of preeclampsia on the offspring. Furthermore, a meta-analysis found no significant increase in fasting blood glucose and insulin levels between offspring exposed in utero to preeclampsia and controls, demonstrating that the risk of developing type 2 diabetes mellitus is unaffected [105]. The limitations of this meta-analysis included a possible causal genetic link between the preeclampsia of the mother and CVD of the offspring, maternal confounders such as maternal BMI and smoking, and heterogeneity among studies. Finally, a very recent meta-analysis by Bi et al. also confirmed the aforementioned absence of an association between in utero exposure to preeclampsia and adverse effects on the lipidemic and glycemic status in offspring under 15 years of age (RR 1.07, 95% CI 0.88–1.32) [106].

### 2.6. Renal System

Renal function impairment represents another significant aspect when analyzing the cardiovascular risks in preeclamptic offspring. Human nephrogenesis occurs mostly in the third trimester when the incidence of preeclampsia is more common, thus having a negative impact on the fetal kidney development [107,108]. Preeclampsia may coexist with FGR and be associated with prematurity, further reducing nephron quantities, decreasing the renal filtration rate, promoting glomerular hypertrophy, and reducing renal vasodilation [109]. The decreased number of nephrons also seems to be independently related to placental dysfunction associated with preeclampsia. The hemodynamic changes and prenatal hypoxia observed in preeclampsia induce molecular, pathophysiological, and histological alterations, which negatively impact fetal renal function and renal vascular tension. Additionally, maternal endothelial dysfunction caused by placental hypoxia leads to an imbalance in maternal vasoactive elements, including increased concentrations of vasoconstrictors (such as thromboxane A2, endothelin, and phenylephrine) and decreased concentrations of vasodilators (such as prostacyclin and nitric oxide) [110,111,112]. This imbalance of vasoactive compounds may lead to reduced vascular relaxation and injury of the endothelium of renal interlobar arteries in fetal kidneys. It has also been suggested that the decrease in the level of vasodilatory nitric oxide in preeclampsia cannot counterbalance the increased sympathetic tone in fetal renal vessels, which is partially induced by the increased sensitivity of the fetus to adenosine, leading to a vasoconstrictive effect [26,27,28]. Finally, early-onset preeclampsia has been associated with an altered renin–angiotensin–aldosterone system in the offspring that persists into adolescence. More specifically, aldosterone levels were found to be elevated in adolescent males born preterm due to preeclampsia. This increase may predispose the offspring of preeclamptic mothers to developing hypertension [89,113].

### 2.7. Endocrine System

Although endocrine diseases in childhood are relatively uncommon, obesity in children and adolescents has developed into a significant public health problem, mainly due to its high prevalence and the association with other comorbidities. It seems that preeclampsia considerably increases the offspring’s risk of long-term endocrine morbidity and, specifically, obesity. The rates of hospital admissions due to endocrine morbidity and obesity for the offspring of preeclamptic pregnancies [98] have been found to be higher compared to controls (0.7% vs. 0.4%; *p* < 0.001 and 0.2% vs. 0.4% *p* < 0.001, respectively), showing a linear correlation with the severity of preeclampsia (0.7% in mild vs. 1.4% in severe preeclampsia, *p* = 0.002). Endocrine morbidity was still significantly increased even after controlling for confounders (OR 1.433 95% CI 1.115–1.841 *p* = 0.005) [98]. The increased obesity prevalence in PE offspring has also been reported in a recent meta-analysis (RR 1.45, 95% confident interval [CI] 1.19–1.78) that further showed that PE offspring were associated with higher mean arterial, systolic, and diastolic blood pressure in puberty. The authors concluded that PE might be associated with central obesity, hypertension, and type 2 diabetes mellitus in offspring later in life [107].

A recent study examined the impact of preeclampsia on the function of the endocrine system in offspring. This was based on the relationship observed between FGR and increased inflammatory markers [114] of leukocytes and CRP in childhood, which, in turn, are associated with insulin sensitivity and obesity [101,115].

A follow-up study [116] of 11-year-old girls and 12-year-old boys showed a remarkable impact of preeclampsia on their androgen levels. More specifically, compared to unexposed female offspring, testosterone levels were much lower in girls born to non-severe preeclamptic pregnancies, while they were higher in a severe-preeclampsia group. In contrast, testosterone levels in boys were higher in all groups of preeclampsia compared to the unexposed group.

Compared to unexposed girls, higher DHEAS concentrations were found in girls exposed to mild and moderate preeclampsia, while lower concentrations were found in a group exposed to severe preeclampsia. The differences in DHEAS concentrations according to the degree of severity of preeclampsia had an impact on the timing of adrenarche, which was probably due to a different androgenic influence. Τhe severe preeclamptic group with lower DHEAS concentrations entered adrenarche relatively late, while earlier menarche and pubarche before thelarche [117] were observed in female offspring exposed to non-severe preeclampsia with a possible increased risk of PCOS and hyperinsulinemia during adulthood [118]^.^ Concerning boys born to severe preeclamptic pregnancies, DHEAS levels were also decreased compared to a control group but did not differ between the mild, moderate, and unexposed groups [117]. Furthermore, boys exposed to mild and moderate preeclampsia also presented increased testicular volume and elevated concentrations of IGF-I, indicating a higher risk of metabolic disorders later in life [118].

Moreover, concerning the long-term reproductive consequences of in utero exposure to hypertensive disorders, these also seem to depend on the gender of the offspring. The timing of pubertal development in male offspring does not seem to be affected by maternal preeclampsia. On the contrary, a recent study proposed a mild acceleration in pubertal timing in the daughters of preeclamptic mothers, while in the daughters of hypertensive mothers, some pubertal milestones seemed to occur earlier than in the daughters of normotensive mothers [119]. Although preeclampsia has been associated with earlier menarche [120,121], some studies found no impact of preeclampsia on the onset of pubertal timing [117,122].

Henley et al. demonstrated a decrease in maternal total and free cortisol concentrations throughout the spectrum of gestational hypertensive disorders. They studied the effect of preeclampsia on the adolescent offspring’s hypothalamic–pituitary–adrenal (HPA) axis function, which revealed mild cases of hypercortisolism, which were possibly caused by an adaptive upregulation of the fetal HPA axis to the reduced maternal cortisol. More specifically, total plasma cortisol, free cortisol, ACTH, and systolic blood pressure were higher in the offspring of preeclamptic pregnancies compared to controls independently of sex and in utero growth. In the long term, this adrenal upregulation may be linked to hypertension, metabolic dysfunction, and neuropsychiatric manifestations [123].

Finally, fetal programming in the offspring of hypertensive mothers may also affect salt sensitivity. A study investigating the different vascular responses of the offspring exposed to experimental late-onset preeclampsia (EPE) found increased aldosterone concentrations in the EPE group with no differences in salt excretion and renal function [124].

In summary, it has been suggested that prenatal exposure to preeclampsia may induce changes in the endocrine system of the offspring, affecting obesity prevalence, adrenal activity, salt sensitivity, androgen balance, and pubertal development.

### 2.8. Respiratory System

#### 2.8.1. Neonatal Life

In preeclamptic offspring, the imbalance of angiogenic and inflammatory factors may dysregulate their pulmonary vascular and alveolar development. Studies on extremely premature (23–28 weeks) [125], premature (30–34+6 weeks) [126], and very-low-birthweight infants [127] have reported an increased risk of severe neonatal respiratory distress syndrome (RDS) (≥30% supplemental oxygen on day 1) [128]. Furthermore, the incidence of neonatal pneumonia, RDS, and lower Apgar scores was found to be higher in the preterm and full-term offspring of pregnancies complicated by gestational hypertensive disorders, for which there was a positive correlation between the severity of maternal hypertension and neonatal respiratory morbidity. Whether in utero exposure to preeclampsia is associated with bronchopulmonary dysplasia (BPD) in preterm infants less than 32 weeks remains controversial. A cohort study [127] of 102 infants has concluded that preeclampsia is not a risk factor for BPD among preterm infants (RR: 0.5, 95% confidence interval [CI]: 0.20–1.20). Nonetheless, other studies [129,130] comparing preterm neonates born to preeclamptic and normotensive mothers have described preeclampsia as a risk factor of BPD, attributing both of these pathologies to abnormal angiogenesis shared in the mother and fetus, while early-onset pulmonary vascular disease in these offspring has been proposed due to the association between the severity of preeclampsia and the severity of BPD.

#### 2.8.2. Childhood

The preeclampsia-mediated effects on systemic and pulmonary circulation seem to be permanent according to a study assessing pulmonary artery pressure and flow-mediated dilation of the branchial artery in children born to preeclamptic women [95]. More specifically, pulmonary artery pressure was 30% higher and flow-mediated dilation was 30% reduced in the offspring of preeclamptic women, while both changes were attributed to augmented oxidative stress related to preeclampsia. Furthermore, the effect of preeclampsia on the relationship between maternal and offspring asthma has been investigated. According to a recent clinical trial [131] comparing the offspring of asthmatic mothers, the risk of asthma was 50% greater for children born to preeclamptic mothers compared to the offspring of normotensive mothers with asthma (adjusted hazard ratio, 2.68; 95% CI: 1.30–5.61). These results support the interaction between already-existing obstetric factors and in utero fetal immune dysregulation because of preeclampsia, making it an independent risk factor for the respiratory morbidity of the offspring [132].

In summary, in utero exposure to preeclampsia may be associated with severe neonatal RDS in addition to asthma and elevated pulmonary artery pressure later in life.

## 3. Conclusions

PE is a syndrome characterized by new-onset hypertension and proteinuria or increased blood pressure and end-organ dysfunction with or without proteinuria after 20 weeks of pregnancy.

The pathophysiology of preeclampsia involves maternal, fetal, and placental parameters. Abnormal placental vasculature development early in pregnancy combined with maternal disease predisposing one to vascular insufficiency may lead to placental hypoperfusion, hypoxia, and ischemia. In turn, this may provoke an excessive release of antiangiogenic factors into maternal circulation, triggering maternal systemic endothelial dysfunction and resulting in hypertension and end organ manifestations.

The multifactorial nature of PE has been confirmed by the overlapping role of genetics and in utero and ex utero environmental aspects such as an unhealthy maternal lifestyle. Additionally, the currently available evidence suggests that there is a link between preeclampsia and adverse effects on offspring’s health. Associations with heart disease, renal dysfunction, general vascular impairment, endocrine disease, immune dysfunction, and neurodevelopmental adverse outcomes have been found. Since hypertensive disorders of pregnancy are commonly related to prematurity and fetal growth restriction, it may not be possible to specify the contribution of each pathology to the increase in the risk for the offspring.

Understanding the mechanisms causing the pathologic changes observed in preeclampsia is crucial for the early prenatal prediction and prevention of PE, the determination of optimal pregnancy follow-up and time of delivery, and the screening of the offspring in jeopardy to ensure early intervention.

## Figures and Tables

**Figure 1 children-10-00826-f001:**
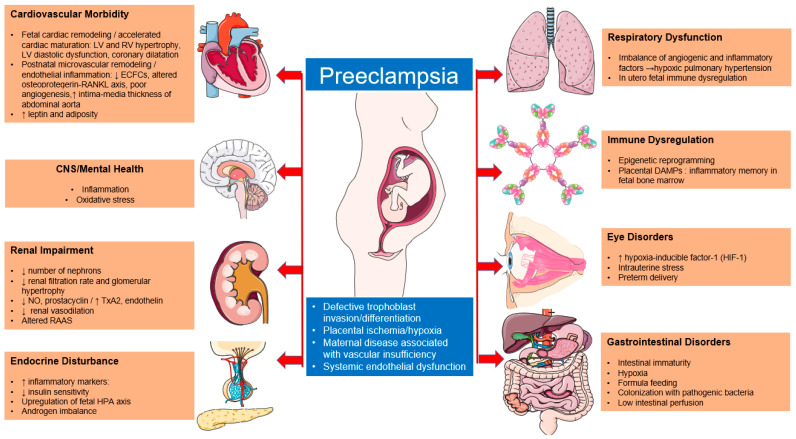
Preeclampsia-associated morbidity in offspring. The Figure was partly generated using Servier Medical Art, a service provided by Servier, licensed under a Creative Commons Attribution 3.0 unported license.

**Table 1 children-10-00826-t001:** Definitions of hypertensive disorders of pregnancy [3,4,5].

	Definition
**Chronic Hypertension**	Systolic BP ≥ 140 mmHg or/and diastolic BP ≥ 90 mmHg on at least two occasions before 20 weeks of pregnancy or hypertension first diagnosed during pregnancy that persists for ≥12 weeks postpartum
**Chronic Hypertension with superimposed preeclampsia**	Development of new-onset proteinuria, other maternal organ dysfunction, or evidence of uteroplacental dysfunction among pregnant women with chronic hypertension
**Gestational Hypertension** **—Nonsevere**	Systolic BP of 140–159 mmHg or/and diastolic BP of 90–109 mmHg on at least two readings 4 h apart after 20 weeks of gestation in a previously normotensive individual
**Gestational Hypertension—Severe**	Systolic BP ≥ 160 mmHg or/and diastolic BP ≥ 110 mmHg on at least two occasions within a short interval (minutes) after 20 weeks of gestation in a previously normotensive individual
**Preeclampsia**	Gestational hypertension (systolic BP ≥ 140 mmHg or/and diastolic BP ≥ 90 mmHg at ≥20 weeks of gestation) accompanied by one or more of the following new-onset conditions at ≥20 weeks of gestation: Proteinuria (≥2+ proteinuria on dipstick test, protein/creatinine ratio ≥ 30 mg/mmol, albumin/creatinine ratio ≥ 8 mg/mmol, or ≥300 mg protein/24 h) Other maternal end-organ dysfunctions, including: ○ Neurologic complications (e.g., eclampsia, altered mental status, blindness, stroke, clonus, severe headaches, or persistent visual scotomata); ○ Pulmonary edema; ○ Hematologic complications (e.g., platelet count 100,000/microL, disseminated intravascular coagulation, hemolysis, etc.); ○ Impaired liver function as indicated by abnormally elevated concentrations of liver enzymes (to more than twice the upper limit normal concentrations) or by severe, persistent right-upper quadrant or epigastric abdominal pain that is unresponsive to medication; ○ Renal insufficiency (Serum Creatinine >1.1 mg/dL or a doubling of the serum creatinine concentration). Uteroplacental dysfunction (e.g., placental abruption, angiogenic imbalance, fetal growth restriction, abnormal results of umbilical artery Doppler waveform analysis, or intrauterine fetal death)
**Preeclampsia with severe features**	Systolic BP ≥ 160 mm Hg or/and diastolic BP ≥ 110 mm Hg at ≥20 weeks of gestation Thrombocytopenia (platelet count < 100,000/microL) Renal insufficiency (Serum Creatinine > 1.1 mg/dL or a doubling of the serum creatinine consecration) Pulmonary edema Impaired liver function as indicated by abnormally elevated concentrations of liver enzymes (to more than twice the upper limit normal concentrations) or by severe, persistent right-upperquadrant or epigastric abdominal pain that is unresponsive to medication Visual disturbances New-onset headaches that are unresponsive to medication
**Eclampsia**	In a patient with PE, a novel onset of tonic clonic, focal, or multifocal seizures in the absence of other causative conditions
**HELLP syndrome**	Hemolysis, elevated concentrations of liver enzymes, and low-platelet syndrome including: Lactate dehydrogenase (LDH) level elevated to 600 IU/L or more; Aspartate aminotransferase (AST) and alanine aminotransferase (ALT) levels elevated to more than twice the upper limit than normal; Platelets count less than 100,000/microL;

**Table 2 children-10-00826-t002:** Short- and long-term outcomes of preeclamptic offspring.

System	Short-Term Outcomes	Long-Term Outcomes
**Neurodevelopmental/CNS ***	■ Cerebral palsy (CP)	■ Autism spectrum disorder (ASD) ■ Attention-deficit/hyperactivity disorder (ADHD) ■ Intellectual disability (ID) ■ Cerebral palsy (CP)
**Cardiovascular**	■ Fetal cardiac remodeling (hypertrophic RV and LV, ventricular dysfunction) ■ Early endothelial inflammation and cardiac cell damage (coronary dilatation, ↑cord NT-proBNP and TnI) ■ Higher SBP, DBP in neonates ■ Congenital Heart Defects	■ Hypertension (higher SDP and DBP) ■ Concentric heart remodeling ■ Ischemic Heart Disease ■ ↑BMI/Obesity
**Renal**	■ Decreased number of nephrons ■ Increased renal vascular tension due to imbalance of vasoactive compounds	■ Altered-RAAS-system-induced increased salt sensitivity
**Endocrine**	■ Upregulation of fetal HPA axis	■ Mild hypercortisolism (increased cortisol levels, ACTH) ■ Androgen (Testosterone, DHEAS) imbalance ■ Accelerated timing of pubarche (girls) ■ Increased IGF-I levels (boys)
**Respiratory**	■ Severe neonatal RDS ■ Bronchopulmonary dysplasia (controversial)	■ Higher pulmonary artery pressure ■ Asthma
**Immune**	■ Neonatal Sepsis	■ Allergic sensitization ■ Severe atopic sensitization
**Gastrointestinal**	■ Necrotizing enterocolitis (NEC)	■ Esophageal morbidity ■ Hernias ■ Functional colonic morbidity
**Eyes**		■ Vascularly associated ophthalmic morbidity

* Neurodevelopmental conditions are typically present at birth; however, diagnosis can be established later in life depending on the severity of the disorder.

## Data Availability

Not applicable.

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
