# Peer review of "Short- and Long-Term Outcomes of Preeclampsia in Offspring: Review of the Literature"

_children, 2023, doi:10.3390/children10050826_

Round 1
Reviewer 1 Report
The paper “Preeclampsia short and long-term outcome in offspring: review of literature.” by Koulouraki et al. describes the review which includes a multifaceted evaluation of the effects of PE on the child. The contents are detailed and considered an important information for many obstetricians; however, there is a large gap among the contents (especially in the level of evidence). Therefore, the contents should not be evaluated in parallel. It is suggested that the overall content needs to be reorganized. All concerns listed below, and the discussion should be rewritten to be more consistent throughout for improving the overall quality.
The content of “Immune system and susceptibility to infections” is superficial and should elaborate on the mechanism of the relationship between infection and preeclampsia. Also, there are few references.
Since “the Cardiovascular System” is detailed, there is an imbalance with the other parts. I think it would be better to remove the articles for which there is little evidence, and to balance the overall presentation. I think it would be better to draw an overall schema and make it easier to understand.
The well-known and important effects of preeclampsia on the children are lifestyle-related diseases, neurodevelopmental disorders, psychosis, and autoimmune diseases, but there are few descriptions of neurodevelopmental disorders and psychosis in this review. The contents with little evidence and low reliability should be deleted or described collectively as things with little evidence.
Author Response
Dear Reviewer,
Thank you for giving us the opportunity to submit a revised draft of our manuscript titled Short and Long-term Outcomes of Preeclampsia In Offspring: Review Of Literature to Children Journal. We appreciate the time and effort that you have dedicated to providing your valuable feedback on our manuscript. We are grateful for your insightful comments on our paper. We have been able to incorporate changes to reflect most of the suggestions provided by you. We have highlighted the changes within the manuscript. Here is a point-by-point response to your comments and concerns.
The paper “Preeclampsia short and long-term outcome in offspring: review of literature.” by Koulouraki et al. describes the review which includes a multifaceted evaluation of the effects of PE on the child. The contents are detailed and considered an important information for many obstetricians; however, there is a large gap among the contents (especially in the level of evidence). Therefore, the contents should not be evaluated in parallel. It is suggested that the overall content needs to be reorganized. All concerns listed below, and the discussion should be rewritten to be more consistent throughout for improving the overall quality.:
Thank you for pointing this out. As recommended, we have modified most of the contents in the neurodevelopmental system and cardiovascular system section to balance these parts. We have also reorganized the endocrinology section (lines 459-473, 520-522) so that it is more easily readable without losing content.
The content of “Immune system and susceptibility to infections” is superficial and should elaborate on the mechanism of the relationship between infection and preeclampsia. Also, there are few references.:
Thank you for your suggestion. We have accordingly revised this section and have added more references, concerning the pathophysiology of preeclampsia impact on the offspring's immune system, on lines 243-258.
Since “the Cardiovascular System” is detailed, there is an imbalance with the other parts. I think it would be better to remove the articles for which there is little evidence and to balance the overall presentation. I think it would be better to draw an overall schema and make it easier to understand.:
We appreciate your insightful comment. Accordingly, we have incorporated a new table with the pathophysiology of preeclampsia for not only the cardiovascular but for all the systems (figure 1, line 105). Additionally, we have removed the articles with little evidence and have merged other articles, so that the different parts be more balanced, and the cardiovascular system part shorter but accurate (lines 310-431).
The well-known and important effects of preeclampsia on the children are lifestyle-related diseases, neurodevelopmental disorders, psychosis, and autoimmune diseases, but there are few descriptions of neurodevelopmental disorders and psychosis in this review. The contents with little evidence and low reliability should be deleted or described collectively as things with little evidence.:
Thank you for your suggestion. We have expanded the pathophysiology in the neurodevelopment section on lines 111-222, adding more references including the also various clinical manifestations of CNS/mental health with their possible link to preeclampsia.
We look forward to hearing from you in due time regarding our submission and to respond to any further questions and comments you may have.
Sincerely,
Sevasti Koulouraki , Vasileios Paschos
Reviewer 2 Report
dear authors
thank you for your submission
we have a few suggestions to improve the quality of the submitted paper
1. overall English polishing is advised
2. many punctuation and fonts errors need carfull revision
3. there were some abbreviations used without defining them, kindly check them
4. many sections were without references of more than 5-7 lines with a single reference. kindly check.
5. Great reviews tacks many citations; it is a good idea to add figures or diagrams to highlight important pathology
6. the rationale behind this review was missed from your abstract
7. keywords could be better improved; check MeSH criteria
8. Mny sections were unbalanced; there were no clear pathophysiology explained; we felt that there were too many opened end
Author Response
Dear Reviewer,
Thank you for giving us the opportunity to submit a revised draft of our manuscript titled Short and Long-term Outcomes of Preeclampsia In Offspring: Review Of Literature to Children Journal. We appreciate the time and effort that you have dedicated to providing your valuable feedback on our manuscript. We are grateful for your insightful comments on our paper. We have been able to incorporate changes to reflect most of the suggestions provided by you. We have highlighted the changes within the manuscript. Here is a point-by-point response to your comments and concerns.
- overall English polishing is advised: Thank you for pointing this out. We agree with this and have incorporated your suggestion throughout the manuscript.
- many punctuation and font errors need careful revision: Thank you for your comment. All spelling and grammatical errors have been reviewed and corrected.
- there were some abbreviations used without defining them, kindly check them: Thank you for your annotation. Accordingly, we have revised the abbreviations, removed them and/or added their definitions throughout the manuscript.
- many sections were without references of more than 5-7 lines with a single reference. kindly check: As suggested we have incorporated more references throughout the text so that no more than five lines in a row remain without a reference.
- Great reviews tack many citations; it is a good idea to add figures or diagrams to highlight important pathology: Thank you for your suggestion. We have, accordingly, added a new figure with the mechanisms of preeclampsia impact on offspring, line 105.
- the rationale behind this review was missed from your abstract: Thank you for pointing this out. We have modified the abstract accordingly so that the aim is clear in the conclusion of it, on lines 22-24.
- keywords could be better improved; check MeSH criteria: We appreciate the feedback and have modified the keywords according to the MeSH criteria on lines 25-26.
- Many sections were unbalanced; there were no clear pathophysiology explained; we felt that there were too many opened end : We agree with this and have incorporated your suggestion throughout the manuscript. More specifically, we have added the aforementioned figure with pathophysiology, and we have added a conclusion of the article (lines 560-582). We have also revised and expanded the pathophysiology and clinical manifestations in the neurodevelopment section on lines 111-222 as well as the immune system section on lines 243-258. Moreover, in order to balance the manuscript, we have merged some of the data in the cardiovascular system (lines 310-431) and endocrine system (lines 459-473, 520-522).
We look forward to hearing from you in due time regarding our submission and to responding to any further questions and comments you may have.
Sincerely,
Sevasti Koulouraki
Vasileios Paschos
Reviewer 3 Report
Thank you for requesting to provide a review of this article, regarding the short and long-term complications of preeclampsia.
The main purpose of the analysis was to study both the short and long-term effects of preeclampsia on the offspring, categorized by system. Many theories of the pathogenesis of preeclampsia were analyzed and yet, its exact ethiology remains unclear and seems to be multifactorial, involving both maternal and placental factors, so the main question adressed in the research was whether the factors that lead to maternal hypertensive disorders, may also cause short and long-term complications on the fetuses and after birth, on the neonats.
The study is a review of literature. The topic is original and relevant in the field and brings usefull knowledge regarding the subject. A comprehensive search strategy was used and so, data from different cohort studies or meta-analyses were evaluated regarding the possible short or long-term outcomes on the offspring. As described in the article, it appears that it is possible that the preeclamptic environment induces epigenetic changes adversely affecting developmental plasticity of the fetus and so, many complications such as neurodevelopmental ones, cardiovacular, renal, endocrine, respiratory, immune etc can be caused. Convincing evidence from meta-analyses show a significant association between preeclampsia and autism spectrum disorders. On the other hand, it appears that the cerebral palsy and the way that preeclampsia may affect these types of disorders is a field of research worth to be studied and so, one large retrospective population-based cohort study showed negative associations between preeclampsia and cerebral palsy for infants born between 23-31 weeks of gestation and positive associations when the delivery is after 37 weeks of pregnancy. The review methodology was comprehensive with screening and data extraction. When it comes to the methodology used, no specific improvements should be considered from my point of view.
The conclusions are consistent with the evidence and the arguments presented, and they adress properly to the main question which conducted the analysis.
The references are appropriate and well suited for this kind of study.
Regarding the figures and pictures used in the article, no figures or tables with the research analysis were used in the review, but being a literature review, such items are not required from my point of view, so no other comments regarding this subject are necessary.
Regarding the structure and accuracy of the phrases, the manuscript has well structured information, with supported evidence and well structured phrases.
The manuscript is original and well defined. The results provide an advance in current knowledge. The results are being interpreted appropriately and are significant, as well as the conclusions.
The study is correctly designed and the analysis is being performed at high standards, so the data are robust enough to draw the conclusion. Surely the paper will attract a wide readership.
To conclude, the article is written in a proper way and brings useful information regarding the subject.
Author Response
Dear Reviewer,
Thank you for giving us the opportunity to submit a revised draft of our manuscript titled Short and Long-term Outcomes of Preeclampsia In Offspring: Review Of Literature to Children Journal. We appreciate the time and effort that you have dedicated to providing your valuable feedback on our manuscript. We are grateful for your insightful comments on our paper. We have been able to incorporate changes to further ameliorate the quality of this manuscript.
More specifically, we have revised and enriched most of the contents in the neurodevelopmental system (lines 111-222), while we have merged data in the cardiovascular system section (lines 310-431) to balance these parts. We have also reorganized the endocrinology section to make it more easily readable without losing content. We have revised the immunity section and have added more references, concerning the pathophysiology of preeclampsia impact on the offspring's immune system, on lines 243-258. Moreover, we have incorporated a new figure with the pathophysiology of preeclampsia for all the systems (line 105). Additionally, we have removed the articles with little evidence. We have also added a conclusion at the end of the article (lines 560-582). Finally, we have revised other punctuation and grammatical mistakes that had skipped our attention.
We look forward to hearing from you in due time regarding our submission and to responding to any further questions and comments you may have.
Sincerely,
Sevasti Koulouraki
Vasileios Paschos
Round 2
Reviewer 1 Report
The authors replied on each comment sincerely and the replies were appropriate. The quality of papers submitted for consideration includes enough reader's interest and scientific quality. The given paper satisfies requirements for publication of this journal.
Reviewer 2 Report
dear editor
the manuscript was revised well
we have o further comment